# Gut Microbiota Manipulation in Irritable Bowel Syndrome

**DOI:** 10.3390/microorganisms10071332

**Published:** 2022-06-30

**Authors:** Tarek Mazzawi

**Affiliations:** 1Faculty of Medicine, Al-Balqa Applied University, Al-Salt 19117, Jordan; tarek.mazzawi@bau.edu.jo; 2Division of Gastroenterology, Department of Medicine, Haukeland University Hospital, 5021 Bergen, Norway

**Keywords:** dysbiosis, FODMAPs, probiotics, antibiotics, fecal microbiota transplantation, enteroendocrine cells

## Abstract

Increased knowledge suggests that disturbed gut microbiota, termed dysbiosis, might promote the development of irritable bowel syndrome (IBS) symptoms. Accordingly, gut microbiota manipulation has evolved in the last decade as a novel treatment strategy in order to improve IBS symptoms. In using different approaches, dietary management stands first in line, including dietary fiber supplements, prebiotics, and probiotics that are shown to change the composition of gut microbiota, fecal short-chain fatty acids and enteroendocrine cells densities and improve IBS symptoms. However, the exact mixture of beneficial bacteria for each individual remains to be identified. Prescribing nonabsorbable antibiotics still needs confirmation, although using rifaximin has been approved for diarrhea-predominant IBS. Fecal microbiota transplantation (FMT) has recently gained a lot of attention, and five out of seven placebo-controlled trials investigating FMT in IBS obtain promising results regarding symptom reduction and gut microbiota manipulation. However, more data, including larger cohorts and studying long-term effects, are needed before FMT can be regarded as a treatment for IBS in clinical practice.

## 1. Introduction

Irritable bowel syndrome (IBS) has become the most common gastrointestinal disease for referral to the gastroenterologists due to the patients’ complaints of abdominal pain, bloating and mixed bowel movements of diarrhea and/or constipation, which can range from mild to severe [1,2]. IBS can cause several extraintestinal symptoms such as headache, tiredness, fibromyalgia and poor social functioning and emotional well-being [3]. Though IBS does not cause increased mortality or cancer incidence [4], the severity of the symptoms reduces the quality of life of the patients, which leads them to skip work or school, reducing their daily productivity and causing a financial burden to society [5,6].

According to a systemic review and meta-analysis [7], the global prevalence of IBS using Rome IV criteria for the diagnosis of IBS and its subtypes [8] is 3.8% [7], but this prevalence was higher in the Western world population using the older criteria Rome III, reaching to 9.2% [7,9]. Diarrhea-predominant IBS is the most common subtype using Rome IV criteria, while mixed-type IBS was the most common subtype using Rome III criteria [7]. IBS is more common in women than in men [7]. The diagnosis of IBS is mainly a diagnosis of exclusion due to the common symptoms that can mask those of other organic diseases, most importantly, celiac disease, inflammatory bowel diseases and colorectal cancer [10].

The cause of IBS is not yet known; however, multiple factors play an important role in the pathogenesis of IBS, such as disturbed gut microbiota (dysbiosis) [11], altered enteroendocrine cells [12,13], previous infections [14,15], genetics [16] and diet [17,18]. Several mechanisms have been suggested for the pathophysiology of IBS, such as alterations in the gut–brain axis [19], abnormalities in the gut endocrine cells and the enteric nervous system [20,21], visceral hypersensitivity [19], gastrointestinal dysmotility [22], postinfectious status and low-grade inflammation [15,23], bacterial overgrowth [24], malabsorption of carbohydrates [25] and altered gut microbiota composition [19,26].

Both pharmacological and non-pharmacological treatments have been used to reduce IBS symptoms [27]. Pharmacological treatment for IBS is mainly symptomatic and of short-term effect [28,29]. Non-pharmacological treatments include dietary management, fecal microbiota transplantation (FMT), psycho- and hypnotherapy and behavioral therapy [27].

In the past decade, research has focused on dietary management and the usage of probiotics, antibiotics and FMT to treat IBS and the current review is an attempt to summarize the effect of these treatment methods on manipulating the gut microbiota.

## 2. Dysbiosis and IBS

The development of healthy gut microbiota essentially begins with its colonization at birth during vaginal delivery with vaginal microbes such as *Lactobacillus*. Delivery using cesarean section results in colonization by skin bacteria or hospital-acquired bacteria, for example, *Staphylococcus* and *Acinetobacter* [30,31], that render these babies susceptible to developing asthma and allergic rhinitis [32]. The gut microbiota matures during the first three years after birth [33] and becomes inhabited by more than 2000 different bacterial species that belong to four main phyla *Bacteroides, Firmicutes, Actinobacteria* and *Proteobacteria* [34]. A healthy composition of the gut microbiota, essential for the proper function of the gastrointestinal tract, is determined by environmental and genetic factors [35]. Some of the environmental factors include dietary habits, geographical location, surgical procedures, smoking, depression, anxiety and recurrent antibiotic treatments [36]. Dysbiosis occurs when an imbalance of the gut microbiota occurs, leading to a reduction in its diversity compared to normal (normobiosis) [37,38] and colonization of opportunistic pathogens [39]. A typical example is what occurs during pseudomembranous colitis when the colon becomes colonized with the opportunistic bacteria *Clostridium difficile* after using broad-spectrum antibiotics, proton pump inhibitor and immunosuppression [39]. Another example is what occurs to patients after a bout of gastroenteritis, causing post-infectious IBS with altered levels of *Bacteroidetes* and *Clostridia* [40]. Decreased richness and diversity of the gut microbiota correlates with increased IBS symptom severity [41] and increased *Firmicutes* to *Bacteroides* ratio as well as increased *Clostridia* and *Clostridiales*, which has been confirmed by a systematic review including 16 articles and involving 777 IBS patients and 461 healthy controls [42]. However, some inconsistencies regarding the microbiota profile of IBS patients exist in the literature. A systematic review of 22 study articles evaluating adults with various IBS symptoms showed that, generally, patients with IBS tend to have decreased levels of *Bifidobacteria* and *Faecalibacterum* (including *Faecalibacterium prausnitzii*) and increased levels of *Lactobacilli* and *Bacteroides* [43], and other studies showed increased levels of *Streptococci* and *Ruminococcus* species when compared to healthy controls [44,45,46]. Previous publications have shown that patients with diarrhea-predominant IBS have lower expression of *Clostridium thermosuccinogenes* phylotype [47], whereas patients with constipation-predominant IBS have increased lactate-producing bacteria that produce sulphide and hydrogen [48].

## 3. Diet

The first-line treatment for IBS has concentrated on changing the patient’s dietary habits [49]. Dietary management has focused on reducing the consumption of certain carbohydrates that are highly fermentable but poorly absorbable, mainly the fermentable oligo-, di-, mono-saccharides and polyols (FODMAPs) that aggravate IBS symptoms, Table 1. When FODMAPs are poorly absorbed in the small intestines, they reach the colon, where they become fermented by the bacteria there, thus producing gas that causes bloating and osmotic changes leading to altered bowel motions [50]. There are at least 10 randomized controlled trials or randomized comparative trials that show that following a low FODMAPs diet improves global IBS symptoms, bloating, flatulence and diarrhea in 50–80% of the patients [51]. Previous publications have shown that changing the type of the consumed diet can change the gut microbiota in these patients with a parallel improvement in their symptoms [44,52,53,54], Table 2.

Several studies showed that using a low FODMAPs diet can reduce the abundance of several bacteria such as *Bifidobacterium*, *Clostridium* and *Faecalibacterium prausnitzii* in feces and increase the richness of Actinobacteria, Firmicutes and Clostridiales [44,53,54], but reported conflicting results concerning other bacterial types and the levels of microbiota metabolites, summarized in Table 2. A study by Staudacher et al. following IBS patients using a low FODMAPs diet for 12 months showed no difference in Bifidobacteria abundance in stool microbiota analysis [55]. However, there were lower concentrations of total fecal short-chain fatty acids, acetate, propionate and butyrate, before and after dietary management [55]. The study concludes that after completing all three phases of the low FODMAPs diet (restriction, reintroduction and personalization), it is safe and effective to follow a low FODMAPs diet for long-term when patients are supervised by a dietician [55]. The responsiveness to a low FODMAPs diet may be predicted by fecal microbiota profiles; for example, *Phascolarctobacterium* are more abundant in responders, and Firmicutes (*Bacilli* and *Clostridia*), *Streptococcus*, *Dorea*, *Coprobacillus* and *Ruminococcus gnavus* are more abundant in non-responders [56]. However, coadministration of multi-strain probiotics preparation (VSL#3) containing *Streptococcus thermophilus, Bifidobacterium breve, B. longum, B. infantis, Lactobacillus acidophilus, L. plantarum, L. paracasei, L. delbrueckii subsp. bulgaricus*, to the low FODMAPs diet restores levels of *Bifidobacterium* species [52].

Dietary fibers play an important role as stool bulking agents as they cause water retention and may increase transit time [57]. They are divided into water-soluble and water-insoluble fibers and consist of short- and long-chain carbohydrates and lignin, Table 1. In IBS, using water-soluble fibers (psyllium) improves IBS symptoms, while water-insoluble fibers exacerbate them [20]. In addition, supplementing one’s diet with fructo- and galacto-oligosaccharides increase the abundance of *Bifidobacterium* due to their prebiotic activity, thus having a beneficial effect on the colon. Moreover, fermenting the dietary fibers by the colonic bacteria leads to the production of fecal short-chain fatty acids (propionate and butyrate) that reflect the activity of the gut microbiota [57,58,59,60].

In studies performed by our group, the aforementioned dietary modifications, using low FODMAPs and changing the dietary fiber intake, also showed significant changes in the densities of the enteroendocrine cells [20]. These cells are scattered throughout the whole gastrointestinal tract and are responsible for releasing the gut hormones that control the functions of the gastrointestinal tract after stimulating their microvilli with different types of nutrients [20]. In biopsies taken from different parts of the gastrointestinal tracts (stomach, small and large intestines) and dyed using special immunohistochemical staining, the densities of the different enteroendocrine cells in IBS patients at baseline were abnormal compared to healthy controls; however, they normalized toward the cells densities measured for healthy controls after dietary modifications [12,20,61,62,63,64,65,66].

## 4. Prebiotics, Probiotics and Antibiotics

Prebiotics, such as trans-galactooligosaccharide, are fermentable poorly-absorbable food elements that provide essential nutrients to enhance the growth of beneficial bacteria, such as *Bifidobacterium* and *Lactobacillus*, and have an anti-inflammatory effect that deters harmful pathogens in the bowels, which may contribute to the improvement of the global symptoms of IBS [67,68]. Probiotics are living microorganisms that consist of bacteria (mainly *Bifidobacterium* and *Lactobacillus*) and yeast, which are friendly to the gut and confer health benefits to the host when given in adequate amounts, usually in tablet forms or consumed in yogurt [58]. Probiotics have been used to beneficially manipulate the dysbiotic gut in IBS patients by improving the function of the gut barrier, inhibiting the overgrowth of pathogenic bacteria and producing short-chain fatty acids and several neurotransmitters [69], normalizing IL-10 and IL-12 levels and suppressing pro-inflammatory cytokine expression [70], Figure 1. Fifty-three randomized–controlled trials involving 5545 patients showed that probiotics appeared to have beneficial effects on global IBS symptoms and abdominal pain [71]. When it comes to the type of probiotics to be used, those that contain multiple bacterial strains (such as a combination of *Bifidobacterium longum*, *B. bifidum, B. lactis, Lactobacillus acidophilus, L. rhamnosus* and *Streptococcus thermophilus*, known as LacClean Gold [72,73] or the combination of seven bacterial strains, namely; *Lactobacillus acidophilus, L. plantarum, L. rhamnosus, Bifidobacterium breve, B. lactis, B. longum and Streptococcus thermophilus* [74], are more beneficial than monostrain probiotics in alleviating IBS symptoms; however, the effects that the probiotics have on improving IBS symptoms are short-termed and do not last for a long time [58,75].

On the other hand, the use of nonabsorbable antibiotics, such as rifixamin, has been investigated in several double-blinded and placebo-controlled randomized controlled trials [71,76] and has been approved by the US Food and Drug Administration for treating patients with diarrhea-predominant IBS [77]. It is suggested that the beneficial effect of using rifixamin in repeated courses occurs by reducing the total load of the gut microbiota and modulating intestinal permeability, thus improving bloating and diarrhea in these patients [78]. The combination of rifixamin and neomycin has improved constipation and bloating in patients with constipation-predominant IBS [79].

## 5. Fecal Microbiota Transplantation

Fecal microbiota transplantation (FMT) has recently become popular as a novel method for modulating gut microbiota in gastrointestinal disorders such as inflammatory bowel syndrome, IBS and recurrent *C. difficile* infection [80,81,82,83], and non-gastrointestinal diseases such as chronic fatigue syndrome, obesity and even some neuropsychiatric disorders [83,84]. During FMT, a suspension made from fecal material, which is collected from healthy individuals, is infused into the gut of the patient via naso-jejunal tube, gastroscope or colonoscope [83], Figure 2. FMT is currently only used in clinical research trials and is considered a safe procedure once one adheres to the current guidelines [85]. Before performing FMT, screening of the donors should include performing thorough physical and laboratory investigations by blood and stool analysis and culture to rule out organic disorders, infectious agents and contagious diseases, most importantly, HIV, viral hepatitis, syphilis, malaria, tuberculosis and trypanosomiasis, to avoid transmitting them to the recipient [86]. It is advisable that the donors of the feces have not recently used antibiotics, travelled to tropical areas, had high-risk sexual behavior or had a bout of gastroenteritis or diarrhea within 4 weeks of donation [85]. It is not yet clear what is the correct dose or the frequency of FMT that should be performed on patients with IBS; however, according to the consensus guidelines, at least 30 g of donor feces should be added to the saline solution in order to prepare the fecal suspension that should be either stored at −80 °C or infusion directly on the same day of preparation [85]. After performing FMT, the stool will be collected from the recipients and stored in a special freezer at −80 °C for further microbial analysis [80]. Table 3 describes the different randomized controlled FMT trials, the dosages and the frequency of fecal transplants.

A recent meta-analysis, including seven placebo-controlled randomized controlled studies, has investigated the effect of this novel method involving 470 patients with IBS [87]. Five out of these trials used fresh/frozen fecal material [88,89,90,91,92], while the other two trials used frozen oral FMT capsules compared to placebo capsules [93,94]. These studies showed conflicting results. The meta-analysis [87] suggested that the form of transplantation used in each study had a significant effect on the study outcome, indicating that the fresh/frozen fecal material might be superior to frozen oral capsules in improving IBS global symptoms and having lasting alteration of gut microbiota, Table 3.

**Table 3 microorganisms-10-01332-t003:** Randomized controlled trials investigating the effect of fecal microbiota transplantation on gut microbiota and microbiota metabolites.

Authors, Years	Diagnostic Criteria, Study Duration	Sample Size, IBS Subtypes	Allocation	Donors	Bowel Cleansing	FMT Route and Location (Upper/Lower GI Tract), Frequency	Dosage of FMT Group	Dosage of Control Group	Microbial Analysis	Findings
Aroniadis et al., 2019 [93]	Rome III, 3 months	*n* = 48: 100% IBS-D	1:1	Four donors, not mixed	No	Oral capsule ( upper), multiple lasted 3 days	25 frozen capsules (0.38 g FMT) per day	25 placebo capsules per day	16S rRNA	Bacterial composition of FMT recipients shifted closer to that of the donors.
El-salhy et al., 2019 [91]	Rome IV, 3 months	*n* = 165: 37.8% IBS-C; 38.4% IBS-D; 23.8% IBS-M	1:1:1	One donor, not mixed	No	Gastroscopy (upper), single FMT	Frozen 30 g FMT and 60 g FMT	Frozen 30 g autologous feces	16S rRNA	Higher abundance of *Eubacterium biforme*, *Lactobacillus* spp. and *Alistipes* spp., lower abundance of *Bacteroides* spp. Inverse correlation between IBS symptoms and the concentrations of *Lactobacillus* spp. and *Alistipes* spp. Negative correlation between the Fatigue Assessment Scale score and the concentration of *Alistipes* spp.
Halkjær et al., 2018 [94]	Rome III, 6 months	*n* = 52: 33.3% IBS-C; 29.4% IBS-D; 37.3% IBS-M	1:1	Four donors, mixed FMT	Yes	Oral capsule (upper), multiple administrations lasted 12 days	25 frozen capsules (50 g FMT)	25 placebo capsules per day	16S rRNA	Fecal donors had higher biodiversity than IBS patients. Microbiota of FMT recipients are more similar to the donors’ microbiota than to that of the placebo recipient. Microbiota of placebo recipient did not become more similar to the donors’ microbiota than patients with IBS before randomization. Bacteroides genus and Ruminococcaceae family correlate positively with IBS symptoms score. Blautia genus and Clostridiales correlate negatively with IBS symptoms score.
Holster et al., 2019 [90]	Rome III, 6 months	*n* = 17: 25% IBS-C; 56.3% IBS-D; 18.8% IBS-M	1:1	Two donors, not mixed	Yes	Colonoscopy (lower), single FMT	Frozen 30 g FMT	Frozen 30 g autologous feces	Human Intestinal Tract Chip (fecal and mucosa)	The abundance of butyrate-producing bacteria in patients’ fecal samples was not lower than the donors at baseline. Microbial composition of patients had changed to resemble that of the donor after FMT. No effect on microbial diversity was observed after FMT in fecal or mucosal microbiota.
Holvoet et al., 2020 [89]	Rome III, 3 months	*n* = 62: 100% IBS-D/IBS-M.	2:1	Two donors; not mixed	No	Naso-jejunal tube (upper), single FMT	Donor fresh feces	Autologous feces	16S rRNA	Donors’ fecal samples had higher diversity than the patients. Responders to FMT had a higher microbial diversity at baseline compared to non-responders. There was a significant difference in overall bacterial composition between responder and non-responders before treatment. Bacterial composition of FMT recipients shifted closer to that of the donors.
Johnsen et al., 2018 [92]	Rome III, 12 months	*n* = 90: 53% IBS-D; 47% IBS-M	2:1	Two donors, mixed	Yes	Colonoscopy (lower), single FMT	Frozen or fresh 50–80 g FMT	Frozen or fresh 50– 80 g autologous feces	Not reported	Not reported
Lahtinen et al., 2020 [88]	Rome III, 3 months	*n* = 55: 51% IBS-D; 14.3% IBS-M; 28.6% IBS unsubtyped; 6.1% other	1:1	One donor, not mixed	Yes	Colonoscopy (lower), single FMT	Frozen 30 g FMT	Fresh 30 g autologous feces	16S rRNA	Changes in gut microbiota profile was observed.

IBS: irritable bowel syndrome; IBS-C: constipation-predominant IBS; IBS-D: diarrhea-predominant IBS; IBS-M: mixed-IBS; FMT: fecal microbiota transplantation; GI: gastrointestinal.

In several studies, butyrate-producing bacteria in fecal samples of the recipient IBS patients were not lower than that of the donors, for example, *Eubacterium halli*, *Eubacterium rectale*, *Megasphera elsdenii*, *Faecalibacterium prausnitzii* [90], *Alistipes* spp. [90,91], *Eubacterium biforme and Lactobacillus* spp. [91] were increased, while *Bacteroides* spp. was decreased in responders following FMT [91]. Moreover, *Lactobacillus* spp. was negatively correlated with the clinical outcome of IBS-symptom severity score [91]. In addition, our group and other publications also reported that IBS patients with low fecal *Alistipes* spp. were most likely to not respond to FMT [41,95] and that FMT also increased the total fecal short-chain fatty acids levels, namely; butyric acid [41,96], which was inversely correlated with IBS symptoms [96]. Donor selection seems to be important, but it remains to be investigated whether single or mixed donors is the preferred choice and at which time intervals should FMT be performed. Several trials showed contradictory results when using either single [91,93,97] or mixed donors [92,94], but most of them showed that the bacterial composition of FMT recipients shifted closer to that of the donors, Table 3. A study showed that increasing the dose of fecal transplant to 60 g and/or using repetitive FMT may increase the response rate in IBS patients to FMT [98].

In several open-labeled studies performed by our group, we investigated the effect of FMT from the healthy relatives of the patients on IBS symptoms, gut microbiota, short-chain fatty acids, stem cells and enteroendocrine cells in diarrhea-predominant IBS patients [41,66,80,99,100]. According to our studies, the IBS symptom severity score improved significantly following FMT [41,80]. The bacterial strains signals for *Ruminococcus gnavus*, *Actinobacteria* and *Bifidobacteria* and the fecal short-chain fatty acids were significantly different between IBS patients and their donors, which became insignificantly different starting at 3 weeks after FMT and lasting up to 6 months following FMT [41]. However, the beneficial effect of FMT on IBS symptoms tends to fade over time, as observed in other trials [41,80,89,91]. The gut microbiota profile for IBS patients became more or less similar to that of their donors following FMT [41,80], which is consistent with the findings of several randomized control trials mentioned in Table 3. Furthermore, our studies also showed that altering the gut microbiota following FMT was paralleled by changes in the densities of the duodenal stem cells progenitors and the densities of the enteroendocrine cells in the duodenum and colon toward the densities measured in healthy controls [66,99,100]. This suggests that manipulating the gut microbiota by FMT changes the so-called “gut microenvironment” in the gastrointestinal tract of IBS patients, which may be responsible for improving the global symptoms of IBS.

## 6. Conclusions

There is strong evidence that dysbiosis plays an important role in the pathophysiology of IBS. There are different non-pharmacological methods that can improve the symptoms of IBS, which affect the gut microenvironment. Manipulating the gut microenvironment not only changes the composition of gut microbiota but also affects the other components of the gut microenvironment, namely short-chain fatty acids that represent the gut microbiota function and the enteroendocrine cells densities with a total impact on the symptoms of IBS. More studies with larger cohorts and for longer terms are required to investigate this issue.

## Figures and Tables

**Figure 1 microorganisms-10-01332-f001:**
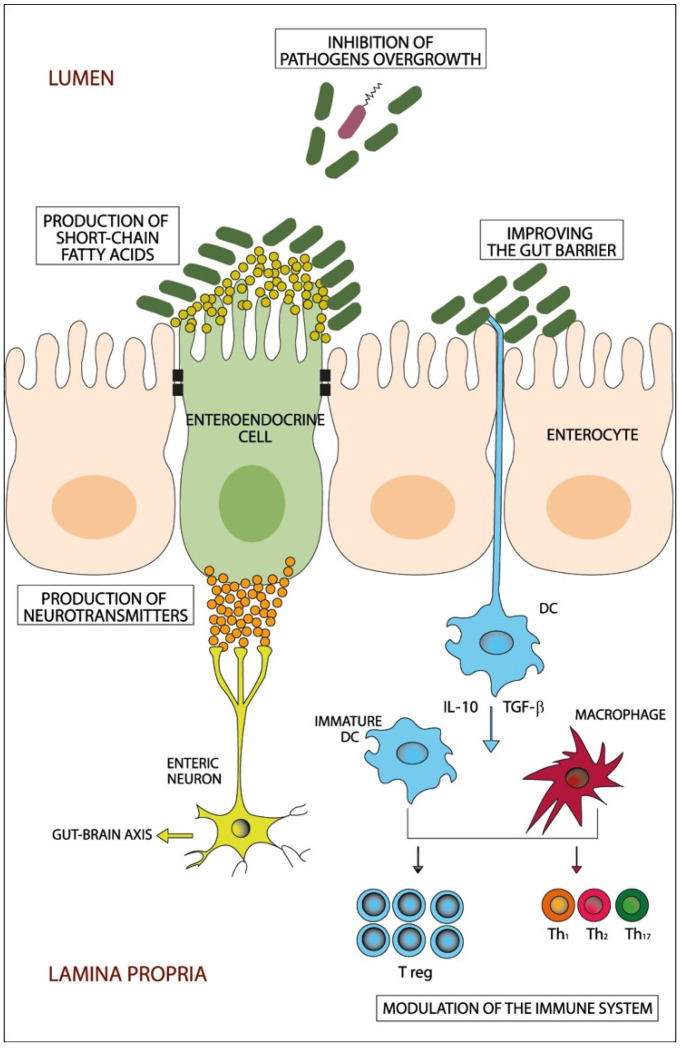
A schematic diagram showing the potential beneficiary effects of probiotics on the gut. Probiotics beneficially manipulate the dysbiotic gut through different potential mechanisms that include inhibition of pathogens’ overgrowth, improving the gut barrier, production of short-chain fatty acids and neurotransmitters and modulation of the immune system. DC: dendritic cells; IL: interleukin; Th: T helper cell; T reg: T regulatory cell; TGF-β: Transforming growth factor-β.

**Figure 2 microorganisms-10-01332-f002:**
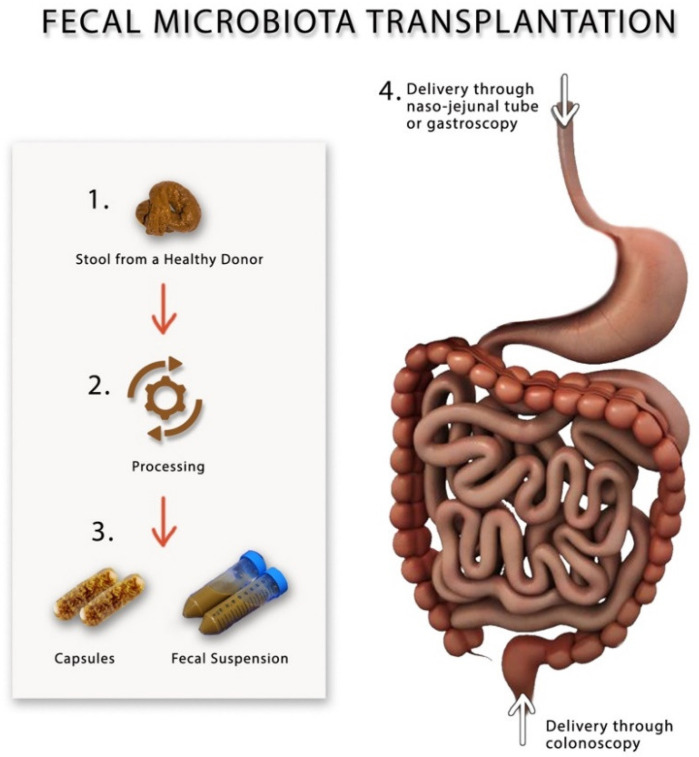
A schematic drawing showing different methods for preparing and performing fecal microbiota transplantation.

**Table 1 microorganisms-10-01332-t001:** Types of FODMAPs and dietary fibers.

FODMAPs	Water Insoluble Fibers	Water Soluble Fibers
Fructose: fruits, honey, corn syrup, agave	Bran	Psyllium
Lactose: milk and dairy	Flax seed	Methylcellulose
Fructans: wheat, onions, garlic	Rye	Calcium polycarbophil
Galactans: legumes (lentils, beans, soybeans)	Non-digestible seeds and vegetables	Inulin
Polyols (sugar alcohols): Xylitol, sorbitol, maltitol, mannitol		Wheat dextrin

**Table 2 microorganisms-10-01332-t002:** Randomized controlled trials investigating the effect of low FODMAP diet on gut microbiota and microbiota metabolites.

Authors, Years	Study Design and Duration	Diagnostic Criteria and Materials	Gut Microbiota		Microbiota Metabolites	
			Microbial Analysis	Findings	Methods	Findings
Halmos EP et al., 2015 [44]	RCT, crossover (single blind), 3 weeks	Rome III IBS and healthy controls. LFD vs. ordinary diet. IBS *n* = 27, Healthy controls *n* = 6	qPCR	Lower absolute abundance of Bifidobacteria, *F. prausnitzii*, Clostridium Cluster IV and lower relative abundance *Akkermansia muciniphila* in LFD than ordinary diet. Lower total bacteria in LFD at baseline. Greater diversity Clostridium Cluster XIV in LFD than ordinary diet at baseline	Gas liquid chromatography	No difference in total or individual stool SCFAs in LFD compared to ordinary diet, baseline.
McIntosh K, et al., 2017 [53]	RCT (single blind), 3 weeks	Rome III IBS. LFD *n* = 19, HFD *n* = 18	16S rRNA sequencing (Illumina)	Higher richness of Actinobacteria, Firmicutes, Clostridiales in LFD than HFD. No difference in α- or β-diversity after LFD vs. baseline. Higher richness in LFD than HFD. Higher abundance of Clostridiales family XIII *Incertae sedis* spp. and *Porphyromonas* spp. in LFD than baseline. Lower abundance of Propionibacteriaceae, Bifidobacteria in LFD than baseline.	Mass spectroscopy	Urinary metabolomic profile at baseline in LFD vs. HFD showed no difference but separated after intervention. Three metabolites (histamine, p-hydroxybenzoic acid and azelaic acid) discriminated groups. Correlations between metabolite concentrations and abundance of various taxa.
Staudacher HM et al., 2012 [54]	RCT (unblind), 4 weeks	Rome III IBS. LFD *n* = 19, Habitual diet *n* = 22	Fluorescence in situ hybridization	Lower abundance of Bifidobacteria in LFD than habitual diet. No difference in total abundance of other groups *(F. prausnitzii*)	Gas liquid chromatography	No difference in total or individual stool SCFAs in LFD compared to habitual diet
Staudacher HM et al., 2017 [52]	RCT (single blind), 4 weeks	Rome III IBS. LFD *n* = 51, Sham *n* = 53	qPCR	Lower abundance of Bifidobacteria in LFD compared to sham	Gas liquid chromatography	Lower stool acetate concentration in LFD compared to control

IBS: irritable bowel syndrome; RCT, randomized controlled trial; LFD, low FODMAP diet; HFD, high FODMAP diet; SCFA, short chain fatty acid; qPCR, quantitative polymerase chain reaction. All differences reported are significant (*p* < 0.05).

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
