# Peer review of "Gut Microbiota Manipulation in Irritable Bowel Syndrome"

_microorganisms, 2022, doi:10.3390/microorganisms10071332_

Round 1

Reviewer 1 Report

The paper of Mazzawi is focused on the manipulation of gut microbiota as novel strategy to improve IBS symptoms. The topic is interesting but the author should deepen the topics covered in each paragraph, in particular, should improve the paragraph concerning the "dysbiosis and IBS" and the "prebiotics, probiotics and antibiotics". In order to improve the manuscript as well as the interest of the reader.

In addition

1.      In paragraph 2. Dysbiosis and IBS

The author should better explain what dysbiosis means, especially in terms of microbiota alteration in favour of the colonization of opportunistic pathogens and should add more information reported in the article cited.

2.       Page 3 Table 1: all the trials listed in table 1 should be discussed in the text 

Page 4:

3.       The author wrote: “the responsiveness to low FODMAPs diet may be predicted by fecal microbiota profiles” (ref 46); what are these profiles?

4.       Immediately after the author wrote: “However, co-administration of probiotics to the low FODMAPs diet restores levels of Bifidobacterium species” (ref42,47); what are the probiotics that are capable to restore Bifidobacterium levels?

5.       Are there any other studies which prove that probiotics improve the symptoms of IBS?

6.       The authors generically talk about probiotics; But what species of probiotics are involved in the cited studies?

Author Response

Dear Editor,

Thank you for the reviewer's comments, we believe they have improved the manuscript. The following is a point-by-point response to the comments.

1. In paragraph 2. Dysbiosis and IBS: The author should better explain what dysbiosis means, especially in terms of microbiota alteration in favour of the colonization of opportunistic pathogens and should add more information reported in the article cited.

Thank you for the comment. Dysbiosis occurs when an imbalance of the gut microbiota occurs leading to  reducetion in its diversity compared to normal (normobiosis) (36,37) and colonisation of opportunistic pathogens (38). A typical example is what occurs during pseudomembranous colitis when the colon becomes colonised with the opportunistic bacteria Clostridium difficile after using broad-spectrum antibiotics, proton pump inhibitor and immunosuppression (38) Another examples is what occurs to patients after a bout of gastroenteritis causing post-infectious IBS with altered levels of Bacteroidetes and Clostridia (39). (Added to text). 

  1. Page 3 Table 1: all the trials listed in table 1 should be discussed in the text: Thank you for the comment. To avoid doubling of information in text and table, a summary of most important findings have been added to the text: Several studies showed that using low FODMAPs diet can reduce the abundance of several bacteria such as Bifidobacterium, Clostridium and Faecalibacterium prausnitzii in feces (43,52,53) but reported conflicting results concerning other bacterial types and the levels of microbiota metabolites, which are summarized in Table 2.

Page 4:

  1. The author wrote: “the responsiveness to low FODMAPs diet may be predicted by fecal microbiota profiles” (ref 46); what are these profiles?  Thank you for the comment. The following has been added to the text: The responsiveness to low FODMAPs diet may be predicted by fecal microbiota profiles, for example, Phascolarctobacterium  are more abundant in responders, and Firmicutes (Bacilli and Clostridia), Streptococcus, Dorea, Coprobacillus and Ruminococcus gnavus are more abundant in non-responders (55). 
  2. Immediately after the author wrote: “However, co-administration of probiotics to the low FODMAPs diet restores levels of Bifidobacterium species” (ref42,47); what are the probiotics that are capable to restore Bifidobacterium levels? Thank you for the comment. The following has been added to the text: However, coadministration of multi-strain probiotics preparation (VSL#3) containing Streptococcus thermophilus, Bifidobacterium breve, B. longum, B. infantis, Lactobacillus acidophilus, L. plantarum, L. paracasei, L. delbrueckii subsp. bulgaricus, to the low FODMAPs diet restores levels of Bifidobacterium species (51).
  3. Are there any other studies which prove that probiotics improve the symptoms of IBS?  Thank you for the comment, references 71 and 72 have been added to the text with examples of the probiotics. 
  4. The authors generically talk about probiotics; But what species of probiotics are involved in the cited studies? Thank you for the comment. The following has been added to the text: When it comes to the type of probiotics to be used, those that contain multiple bacterial strains (such as a combination of Bifidobacterium longum, B. bifidum, B. lactis, Lactobacillus acidophilus, L. rhamnosus and Streptococcus thermophiles, known as LacClean Gold (71,72) or the combination of seven bacterial strains, namely; Lactobacillus acidophilus, Lactobacillus plantarum, Lactobacillus rhamnosus, Bifidobacterium breve, Bifidobacterium lactis, Bifidobacterium longum, and Streptococcus thermophilus (73).

Reviewer 2 Report

I highlight below where I found a few misspellings, and make several suggestions where, in my opinion, the content (especially figures) of the article can be adjusted slightly to help the novice reader. I also point out one area - pathogen pre-screening of FMT-donor stool samples - where the author may want to add some content to the review. Overall, this is an excellent review article that is well written and easy to understand the content, and only needs a few minor tweaks before publication.

Comments by Section:

Diet:

Lines 75-77: It might be useful to include a table or a small figure the highlights some of the specific carbohydrates that are classified as FODMAPs – most tables online list foods that should be avoided rather than listing the specific carbohydrates. One simple route would be to get copyright permission to share an existing figure from a publication.

 Prebiotics, Probiotics, and Antibiotics:

Line 127: there seems to be a misspelling. Consider changing “youghurt” to either “yoghurt (non-American English spelling)” or “yogurt (American English spelling).”

Line 135: there seems to be a misspelling. Consider changing “allevisting” to “alleviating.”

 Fecal Microbiota Transplantation:

Figure 1, Line 152: When a solid material – such as donor feces in an FMT procedure - is dispersed in a liquid media, it is known as a dispersion or suspension. Consider changing “Liquid Solution” to “Fecal Dispersion” or “Fecal Suspension.”

 Line 158: there seems to be a misspelling. Consider changing “FMT capsles” to “FMT capsules.” Double check spelling throughout the entire manuscript.

Questions by Section:

Introduction:

Lines 28-32: Would it be worthwhile to a reader entering the field to add a sentence or two to summarize the existence or lack of established links between IBS and colorectal cancer? Is it possible that a diagnosis of IBS can mask a colon cancer diagnosis? There is a rich body of literature on this topic that can be summarized briefly here in the intro to add a bit more depth to this area of research.

Prebiotics, Probiotics, and Antibiotics:

Lines 127-131: Would a small figure outlining the beneficial effects conferred by probiotic consumption help the reader? It seems like it would be useful to show microbes labeled as probiotic organisms residing in the intestinal tract that add bulk to the mucosal layer, suppress pathogenic organisms, normalize immune compounds (IL-10 and IL-12) and suppress pro-inflammatory cytokines, and increase intestinal neurotransmitters and SCFA production. A picture is worth a thousand words, and this might be a great addition to this manuscript.

Fecal Microbiota Transplantation:

Figure 1, Line 152: Would it be beneficial to illustrate what a typical dosing schedule for FMT in IBS looks like (number of doses over time)? Is there any clinical microbiology performed to assess microbiome shifts in response to FMT administration? How many days typically offset between a dose and the clinical microbiology assessment? A small inset in the figure would help a novice reader visualize the overall therapeutic regimen.

 Lines 165-179, or Lines 180-197: Should the current state of the art in pathogenic screening of FMT-donor stool be described here? This is an important consideration that should at least briefly be mentioned in this article based on the recent issues with ESBL-producing E coli infections post-FMT (DeFilipp Z et al., N Engl J Med, 2019, 381:2043-2050).

Author Response

Thank you for the reviewer's comments which we believe have improved the manuscript. 

The followings are point-by-point responses:

Diet:

Lines 75-77: It might be useful to include a table or a small figure the highlights some of the specific carbohydrates that are classified as FODMAPs – most tables online list foods that should be avoided rather than listing the specific carbohydrates. One simple route would be to get copyright permission to share an existing figure from a publication.

Thank you for the comment. Table 1 has been constructed with both FODMAPs and types of dietary fibers. 

 Prebiotics, Probiotics, and Antibiotics:

Line 127: there seems to be a misspelling. Consider changing “youghurt” to either “yoghurt (non-American English spelling)” or “yogurt (American English spelling).” 

Thank you for the comment. Spelling mistake is checked.

Line 135: there seems to be a misspelling. Consider changing “allevisting” to “alleviating.”

Thank you for the comment. Spelling mistake is checked.

 Fecal Microbiota Transplantation:

Figure 1, Line 152: When a solid material – such as donor feces in an FMT procedure - is dispersed in a liquid media, it is known as a dispersion or suspension. Consider changing “Liquid Solution” to “Fecal Dispersion” or “Fecal Suspension.”

Thank you for the comment. Liquid solution has been changed to fecal suspension in figure 1.

 Line 158: there seems to be a misspelling. Consider changing “FMT capsles” to “FMT capsules.” Double check spelling throughout the entire manuscript.

Thank you for the comment. Spelling mistake is checked.

Questions by Section:

Introduction:

Lines 28-32: Would it be worthwhile to a reader entering the field to add a sentence or two to summarize the existence or lack of established links between IBS and colorectal cancer? Is it possible that a diagnosis of IBS can mask a colon cancer diagnosis? There is a rich body of literature on this topic that can be summarized briefly here in the intro to add a bit more depth to this area of research.

The following text has been added: The diagnosis of IBS is mainly a diagnosis of exclusion due to the common symptoms that can mask those of other organic diseases; most importantly, celiac disease, inflammatory bowel disease and colorectal cancer (9).

Prebiotics, Probiotics, and Antibiotics:

Lines 127-131: Would a small figure outlining the beneficial effects conferred by probiotic consumption help the reader? It seems like it would be useful to show microbes labeled as probiotic organisms residing in the intestinal tract that add bulk to the mucosal layer, suppress pathogenic organisms, normalize immune compounds (IL-10 and IL-12) and suppress pro-inflammatory cytokines, and increase intestinal neurotransmitters and SCFA production. A picture is worth a thousand words, and this might be a great addition to this manuscript.:

Thank you for the suggestion which has been added as figure 1.

Fecal Microbiota Transplantation:

Figure 1, Line 152: Would it be beneficial to illustrate what a typical dosing schedule for FMT in IBS looks like (number of doses over time)? Is there any clinical microbiology performed to assess microbiome shifts in response to FMT administration? How many days typically offset between a dose and the clinical microbiology assessment? A small inset in the figure would help a novice reader visualize the overall therapeutic regimen.:

Thank you for the comment. FMT for IBS is currently used only in clinical research. No clear guidelines currently exist about the number of doses or time intervals  (added to text). Microbial analysis results are mentioned both in text and table 3. 

 Lines 165-179, or Lines 180-197: Should the current state of the art in pathogenic screening of FMT-donor stool be described here? This is an important consideration that should at least briefly be mentioned in this article based on the recent issues with ESBL-producing E coli infections post-FMT (DeFilipp Z et al., N Engl J Med, 2019, 381:2043-2050).

Thank you for the comment. The following has been added to the text which summarizes all the above questions including this one: 

FMT is currently only used in clinical research trials and is considered as a safe procedure once one adheres to the current guidelines (84). Before performing FMT, screening of the donors should include performing thorough physical and laboratory investigations by blood and stool analysis and culture to rule out organic disorders, infectious agents and contagious diseases most importantly; HIV, viral hepatitis, syphilis, malaria, tuberculosis and trypanosomiasis, to avoid transmitting them to the recipient (85). It is advisable that the donors of the feces have not used antibiotics, recently travelled to tropical areas, had sexual high-risk behaviour or had a bout of gastroenteritis or diarrhea within 4 weeks of donation (84). It is not yet clear what is the correct dose or the frequency of FMT that should be performed to patients with IBS, however according to the consensus guidelines at least 30 g of donor feces should be added to saline solution in order to prepare the fecal suspension that should be either stored at -80℃ or infusion directly on the same day of preparation (84). After performing FMT, stool will be collected from the recipients and stored in special freezer at -80℃ for further microbial analysis (79). Table 3 describes the different randomized controlled FMT trials, the dosages and frequency of fecal transplants.

Round 2

Reviewer 1 Report

The Author  replied to all comments and questions asked. However, the tables should be improved in graphic and formatting.